# LightMotion: A Light and Tuning-free Method for Simulating Camera Motion in Video Generation

## Abstract

Existing camera-controlled video generation methods face computational bottlenecks, either due to significant fine-tuning overhead or heavy inference processes. In this paper, we proposes LightMotion, a light and tuning-free method for simulating camera motion in video generation. Operating in the latent space, it eliminates additional fine-tuning, inpainting, and depth estimation, making it more streamlined than existing methods. The endeavors of this paper comprise: (i) The latent space permutation operation simulates three basic camera motions: panning, zooming, and rotation, whose combinations cover almost all real-world movements. (ii) The latent space resampling strategy combines background-aware sampling with cross-frame alignment, accurately filling new perspectives while maintaining coherence across frames. (iii) Our analysis reveals that the tuning-free permutation and resampling will cause an SNR shift in latent space, leading to poor-quality generation. To address this, we propose the latent space correction scheme, which mitigates the shift and consequently improves video quality. Extensive experiments validate the superiority of LightMotion over other baselines.

## 1 Introduction

Sora Liu et al. (2024) has fostered the development of numerous open-source video generation frameworks Wang et al. (2023); Blattmann et al. (2023a); Chen et al. (2023), accelerating progress in video generation field. Building on these frameworks, early studies focus on controllable video generation, which conditions on pose maps Hu (2024), style images Song et al. (2024), and depth maps Wang et al. (2024b), thereby achieving more precise results. However, these approaches typically lack explicit camera controls, which remains a limitation in real world. With the growing demand for flexible video generation in virtual reality, this limitation becomes particularly critical.

In response, numerous studies Wang et al. (2024d); Xu et al. (2024); Zheng et al. (2024); Chen et al. (2023) have recently begun exploring camera-controllable video generation approaches, alleviating industry needs. Although these methods show remarkable effectiveness, they still encounter three main challenges: *(i) Expensive fine-tuning overheads.* To achieve camera control, methods such as CameraCtrl He et al. (2025) fine-tune an encoder conditioned on camera parameters, based on the generic video diffusion model AnimateDiff Guo et al. (2024). However, this process requires up to 400 A100-80GB GPU hours, and the heavy computational cost together with limited data makes these methods impractical for broad adoption. *(ii) End-to-end inference bottlenecks.* To optimize computational resources, CamTrol Hou et al. (2024) employs a depth estimation model Bhat et al. (2023) for view warping and utilizes an inpainting model Rombach et al. (2022) to generate novel views, all in a tuning-free manner. While effective, these additional models increase inference complexity, ultimately making end-to-end inference infeasible. *(iii) Restricted camera motions.* As shown in Figure 1(a), generic video generation models like Animatediff Guo et al. (2024) typically produce outputs from a fixed perspective. MotionBooth Wu et al. (2024) mitigates this limitation by shifting latent-space pixels across frames to simulate simple camera panning without additional fine-tuning. However, it remains unable to capture general real-world camera motions such as zooming and rotation. Therefore, developing a lightweight, tuning-free, and end-to-end video generation framework for general camera motions remains a significant challenge.

In this paper, we propose **LightMotion**, a light and tuning-free approach for simulating general camera motion in video generation. To enable general camera motion, we introduce **latent space permutation** during denoising, which alters the relative ordering of pixels in latent space. The latent space permutation is composed of three fundamental types of camera motion: *(i) Panning in the x-y plane:* shifting latent-space pixels frame-by-frame to simulate horizontal and vertical movement; *(ii) Zooming along the z axis:* interpolating latent-space pixels to simulate perspective scaling; *(iii) Rotation around the x, y, and z axes:* extending point cloud projection theory Chen et al. (2023) to latent space for rotation, which is proven to be depth-independent, thereby eliminating additional depth estimation models. The combination of these three basic motions can represent almost all real-world camera movements.

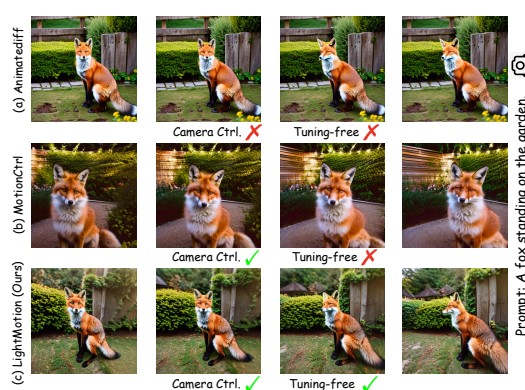

Figure 1: Comparisons with existing methods. (a) Animatediff Guo et al. (2024) produces fixed-viewpoint videos. (b) MotionCtrl Wang et al. (2024d) fine-tunes Animatediff to achieve camera control. (c) LightMotion allows Animatediff to simulate camera motions without fine-tuning.

The original latent is updated through permutation, effectively simulating various camera movements. However, some positions in the updated latent lack corresponding values due to the emergence of new perspective. MotionBooth Wu et al. (2024) addresses this by sampling pixels from the original latent based on semantic similarity. However, incorrect object region sampling and frame-independent sampling often lead to object duplication and inconsistent artifacts, as shown in Figure 3. To address these issues, we propose **latent space resampling**, applied after latent space permutation. It incorporates a *background-aware sampling* strategy that leverages cross-attention maps to restrict sampling to background regions, and a *cross-frame alignment* mechanism that propagates sampled values across frames for better temporal consistency.

Our experiments further reveal that the tuning-free update operation often degrades video quality, particularly in general camera motion scenarios. This limitation is overlooked by MotionBooth Wu et al. (2024), as it only considers simple camera panning. To investigate this phenomenon, we conduct an in-depth analysis from an signal-to-noise ratio (SNR) perspective. The results show that the update operation induces an SNR shift in the latent space, which creates a gap between training and inference, ultimately degrading generation quality. To address this, we propose **latent space correction**, which reintroduces noise into the latent space to uniformly correct the SNR shift and alleviate corresponding gap. Experimental results demonstrate that our LightMotion (Figure 1(c)) produces camera motion comparable to the tuning-based MotionCtrl Wang et al. (2024d) (Figure 1(b)).

## 2 RELATED WORKS

**Camera Motion Video Generation via Tuning.** Video generation with camera motion has recently gained significant research interest. Early works mainly focus on fine-tuning specific datasets with camera motion. MotionCtrl Wang et al. (2024d) and CameraCtrl He et al. (2025) train an encoder to process camera parameters and inject encoded features into temporal attention layers for perspective control. CamCo Xu et al. (2024) and CamI2V Zheng et al. (2024) leverage camera parameters to compute epipolar lines across different perspectives, constructing masks to constrain frame-to-frame attention, thereby improving the modeling of physical scene information. ViewCrafter Chen et al. (2023) first estimates the depth map of an image and projects this image into 3D point cloud. These point clouds are projected into different perspectives, with the missing regions filled in by an additional fine-tuned inpainting model.

**Tuning-free Camera Motion Video Generation.** The heavy fine-tuning burden of video generation models and the scarcity of specialized datasets restrict the development of tuning-based methods. Consequently, recent works attempt to enable base models Sterling (2023); Guo et al. (2024); Wang et al. (2024c) to perform camera-controllable video generation in a tuning-free man-

ner. CamTrol Hou et al. (2024) leverages a depth estimation model Bhat et al. (2023) to generate a 3D point cloud from an image, renders it from user-defined perspective, and uses an inpainting model Rombach et al. (2022) to fill gaps caused by the perspectives transformation. The most related work is MotionBooth Wu et al. (2024), which proposes a latent-shift operation to simulate simple camera panning. However, it suffers from object duplication and inconsistent artifacts due to random frame-by-frame sampling. Key advances of our LightMotion over other tuning-free methods include: (i) no need for additional depth estimation or inpainting models; (ii) latent space permutation operation that supports general camera motions beyond simple panning; (iii) background-aware sampling and cross-frame alignment to address object duplication and inconsistent artifacts; (iv) latent space correction to mitigate SNR shift and quality degradation caused by latent updates.

## 3 PRELIMINARY

### 3.1 LATENT VIDEO DIFFUSION MODEL

The Latent Video Diffusion Model Blattmann et al. (2023b) uses an encoder $\mathcal{E}$ to map $N$ frames of video $I^{1:N}$ to the latent space $Z_1^{1:N}$. During training, $Z_1^{1:N}$ is noised through:

$$Z_t^{1:N} = \sqrt{\bar{\alpha}_t} \cdot Z_1^{1:N} + \sqrt{1 - \bar{\alpha}_t} \cdot \varepsilon^{1:N}, \quad \varepsilon^{1:N} \sim \mathcal{N}(0, I), \tag{1}$$

where $\{\alpha_t\}_{t=1}^T$ represents a predefined variance schedule, and $\bar{\alpha}_t$ is defined as $\prod_{i=1}^t \alpha_i$.

Given the noised latent $Z_t^{1:N}$, a neural network (e.g., U-Net) predicts the added noise $\hat{\epsilon}_t$, supervised by the mean squared error (MSE) loss and guided by the text prompt $y$: Upon completion of training, the network will start with a standard Gaussian noise $Z_T^{1:N}$, progressively predicts the noise $\varepsilon_\theta^{1:N}$, and executes the DDIM Song et al. (2021) schedule as follows:

$$Z_{t-1}^{1:N} = \sqrt{\bar{\alpha}_{t-1}} \left( \frac{Z_t^{1:N} - \sqrt{1 - \bar{\alpha}_t} \cdot \varepsilon_\theta^{1:N}}{\sqrt{\bar{\alpha}_t}} \right) + \sqrt{1 - \bar{\alpha}_{t-1}} \cdot \varepsilon_\theta^{1:N}. \tag{2}$$

Finally, the decoder $\mathcal{D}$ apply the operation $I'^{1:N} = \mathcal{D}(Z_1^{1:N})$ to reconstructs the final video $I'^{1:N}$.

### 3.2 CROSS-ATTENTION MAP

In cross-attention layers, the intermediate feature $\phi(Z_t)$ is mapped to $Q$ via the learnable matrix $W_Q$. Meanwhile, the text prompt $y$ is encoded by $\tau$ and mapped to $K$ and $V$ via the learnable matrices $W_K$ and $W_V$. The attention process between the text and intermediate features is as follows:

$$\mathcal{A} = \text{Softmax}\left(\frac{Q \cdot K}{\sqrt{d}}\right), \quad \text{Attn}(Q, K, V) = \mathcal{A} \cdot V, \tag{3}$$

where $d$ is the dimension of $Q$ and $K$, $\mathcal{A} \in \mathbb{R}^{N \times (hw) \times L}$ is cross-attention map, with $hw$ and $L$ denoting the number of visual and text tokens. Higher values in $\mathcal{A}$ reflect stronger correlation between the text token and generated region.

## 4 METHOD

In this section, we propose LightMotion, a light and tuning-free method for simulating camera motion in video generation. The overall pipeline is illustrated in Figure 2. Starting from a Gaussian noise $Z_T^{1:N}$, LightMotion first denoises it until timestep $T_0$, as formulated in Eq. (2). Then, the latent code $Z_{T_0}^{1:N}$ is updated to $Z_{T_0}'^{1:N}$ through *latent space permutation* and *latent space resampling*. Next, LightMotion continues denoising until timestep $T_1$ to preserve the semantics integrity and camera motion. Subsequently, it performs *latent space correction* to reintroduce noise from $Z_{T_1}'^{1:N}$ to $Z_{T_2}^{*1:N}$ using Eq. (1), mitigating the SNR shift caused by the tuning-free update operation. Finally, LightMotion completes the remaining denoising from timestep $T_2$ to 1.

### 4.1 LATENT SPACE PERMUTATION

To address the limitation of MotionBooth Wu et al. (2024), which only considers simple panning, we present a generalized motion formulation that captures various camera movements. Recall from

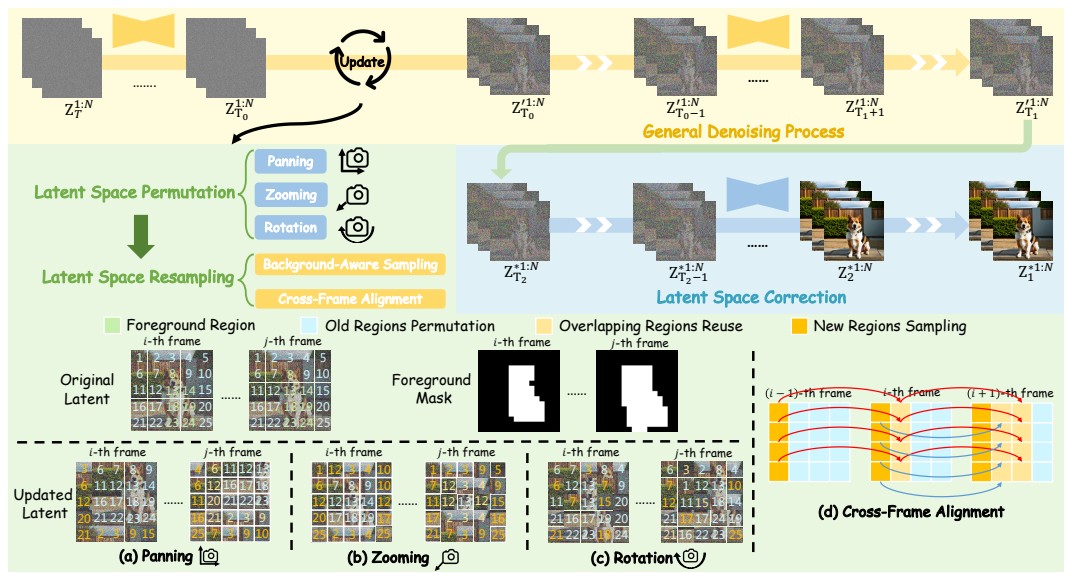

Figure 2: The overall pipeline of LightMotion, which simulates camera motion in video generation without additional fine-tuning. (a–c) illustrate permutation and resampling in latent space for panning, zooming, and rotation, respectively. Numbers denote positions rearranged by camera motion. (d) shows a toy example of the cross-frame alignment strategy.

Figure 1 that generic video generation models typically operate from a fixed perspective. To approximate general camera motion, we design a mapping function $\mathcal{F}$ that permutes the original latent representations on a frame-by-frame basis. Given the pixel at coordinates $[u, v, 1]$ in the original latent $Z_{T_0}^{1:N}$, $\mathcal{F}$ maps it to new coordinates $[u', v', 1]$, where the 1 denotes the homogeneous coordinate. The updated latent $Z_{T_0}'^{1:N}$ is then constructed by copying original value. As described below:

$$Z_{T_0}'^{1:N}[u', v', 1] = Z_{T_0}^{1:N}[u, v, 1], \tag{4}$$

$$[u', v', 1]^T = \mathcal{F}^{1:N}([u, v, 1]^T), \tag{5}$$

where the new coordinates $[u', v', 1]$ are discarded if they exceed the latent space dimensions $h$ or $w$, as these coordinates fall outside the corresponding camera perspective.

We decompose the general mapping function $\mathcal{F}^{1:N}$ into three fundamental geometric components: panning in the x–y plane ($\mathcal{F}_{panning}^{1:N}$), zooming along the z-axis ($\mathcal{F}_{zooming}^{1:N}$), and rotation around the x, y, and z axes ($\mathcal{F}_{rotation}^{1:N}$). Combinations of these functions can effectively represent general camera motions.[1] Examples are shown in Fig. 2(a–c), with coordinates rounded omitted for clarity.

**Panning.** Camera panning shifts the relative order of pixels in the latent space along the panning direction, and the coordinate mapping for the $i$-th frame is defined as:

$$\mathcal{F}_{paninig}^{i}([u', v', 1]^T) := \begin{bmatrix} 1 & 0 & \frac{x \cdot w \cdot i}{N} \\ 0 & 1 & \frac{y \cdot h \cdot i}{N} \\ 0 & 0 & 1 \end{bmatrix} \cdot \begin{bmatrix} u \\ v \\ 1 \end{bmatrix}, \tag{6}$$

where $x, y \in [-1, 1]$ are user-specified panning parameters following Yang et al. (2024), which respectively define the movement ranges along the $X$ and $Y$ axes as relative proportions of the width $w$ and height $h$ in the latent space.

**Zooming.** Camera zooming naturally involves a change in perspective through spatial scaling. We simulate this effect by interpolating the latent representation at each frame, with the coordinate mapping for the $i$-th frame defined as:

$$\mathcal{F}_{zooming}^{i}([u', v', 1]^T) := \mathcal{T}([u, v, 1]^T, s^i), \tag{7}$$

---

[1]Detailed proofs are provided in the **supplementary materials**.

where $\mathcal{T}$ is the interpolation transformation, and the scaling factor $s^i$ for the $i$-th frame is $1 + \frac{z \cdot i}{N}$, with $z \in [-1, 1]$ as the user-specified zooming parameter like Yang et al. (2024).

**Rotation.** Camera rotation in latent space mirrors its behavior in pixel space. Therefore, we extend the point cloud projection theory Hou et al. (2024) into latent space. Only rotation is considered, as translation is included by panning and zooming. The coordinate mapping for the $i$-th frame is:

$$\mathcal{F}^i_{rotation}([u', v', 1]) := K \cdot R^i \cdot K^{-1} \cdot \begin{bmatrix} u \\ v \\ 1 \end{bmatrix} \cdot d(u, v, 1), \tag{8}$$

where $K$ is the camera's intrinsic matrix, $R^i$ is the rotation matrix for the $i$-th frame that associated by user-specified rotation parameter $\theta^2$, and $d(u, v, 1)$ is associated depth information. When rotation is limited to the x-y-z axes, the rotation coordinate projection is independent of $d(u, v, 1)^3$, which inherently bypasses depth estimation requirements.

## 4.2 LATENT SPACE RESAMPLING

While latent space permutation effectively simulates various camera motions, it leaves certain positions of the updated latent $Z'^{1:N}_{T_0}$ without corresponding values due to *the emergence of new perspective*. To address this, Motion-Booth Wu et al. (2024) fills these positions of new perspective by randomly sampling pixels from the original latent frame-by-frame. Whereas, as shown in Figure 3(a), this strategy often produces repetitive objects that conflict with the original prompt. Moreover, the per-frame independent sampling leads to artifacts, as illustrated in Figure 3(b). To overcome the issues above, we propose the *background-aware sampling* for more accurate filling, coupled with the *cross-frame alignment* to ensure coherence across frames.

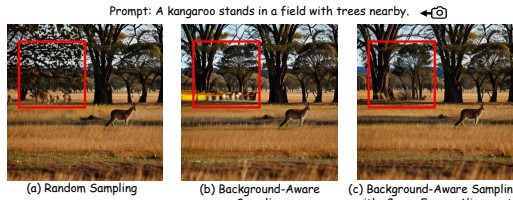

Figure 3: Comparison of different methods. (a) Random sampling causes object repetition. (b) Frame-by-frame sampling introduces artifacts due to temporal inconsistency. (c) Our method produces accurate results without artifacts.

**Background-Aware Sampling.** Through empirical experiment, we observe that repetitive generation arises from random sampling, which incorrectly samples new pixels from the foreground region of the original latent. To mitigate this, we use the cross-attention map $\mathcal{A} \in \mathbb{R}^{N \times (hw) \times L}$ formulated in Eq. (3) to dynamically locate background regions during denoising, effectively avoiding the need for segmentation models. Specifically, we first extract $\mathcal{A}$ from the last upsampling block in U-Net, during the preceding denoising process (from $T$ to $T_0$). Then, $\mathcal{A}$ is further averaged, binarized, and refined by erosion and dilation Lin et al. (2024; 2025). Next, we select the subject (foreground) token via the Dependency Parsing Honnibal (2017) and use its activation regions in $\mathcal{A}$ as mask $\mathcal{M} \in \mathbb{R}^{N \times h \times w}$, where 0 denotes background region. Finally, new pixels at $[j, k]$ in the updated latent $Z'^{1:N}_t$ are sampled from rows or columns of the original latent's background region:

$$\begin{aligned} Z'^{1:N}_{T_0}[j, k] &= Z^{1:N}_t[j, l] \quad s.t. \ \mathcal{M}^{1:N}[j, l] = 0, \\ {\scriptstyle (j,k) \in \Omega^{1:N}} \\ Z'^{1:N}_{T_0}[j, k] &= Z^{1:N}_t[l, k] \quad s.t. \ \mathcal{M}^{1:N}[l, k] = 0, \\ {\scriptstyle (j,k) \in \Omega^{1:N}} \end{aligned} \tag{9}$$

where $\Omega^{1:N}$ represents the new regions across frames in the updated latent that require resampling.

**Cross-Frame Alignment.** While background-aware sampling alleviates repetitive generation, frame-wise sampling introduces artifacts, as shown in Figure 3(b). In practice, continuous camera motion creates overlapping perspectives across frames, which require consistent pixel values. However, applying Eq. (9) to fill these regions independently for each frame leads to inconsistencies

---

[2]See the **supplementary materials** for detailed definitions.

[3]See the **supplementary materials** for detailed proof.

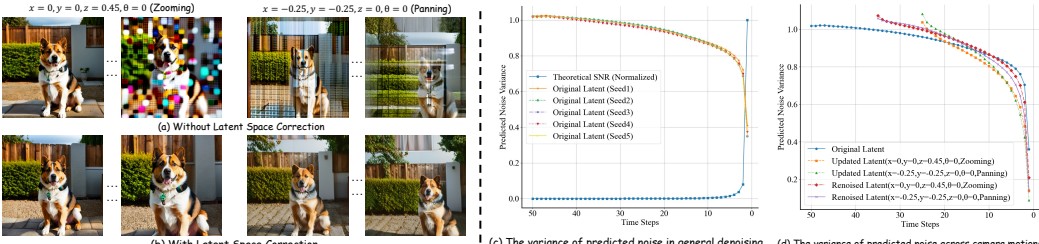

Figure 4: (a) The update operation with different camera motions results in poor-quality outputs. (b) Latent space correction significantly enhances the quality of generated videos. The results are averaged over 1,000 samples for reliability: (c) The variance of predicted noise is negatively correlated with the SNR of input latent. (d) The update operation alters the variance of predicted noise, while our correction mechanism effectively mitigates this deviation.

and visible artifacts. To maintain coherence, we reuse sampled results from the previous frame in overlapping regions, as illustrated in Figure 2(d). Specifically, when filling the updated latent $Z_{T_0}^{'i}$, we first copy pixels from overlapping regions $(j, k) \in \mathcal{H}^{(i,i-1)}$ in $Z_{T_0}^{'i-1}$, as formulated below:

$$Z_{T_0}^{'i}[j,k] \underset{(j,k)\in\mathcal{H}^{(i,i-1)}}{=} Z_{T_0}^{'i-1}[\mathcal{G}_{i\to i-1}(j,k)], \qquad (10)$$

where $\mathcal{G}_{i\to i-1}$ maps pixel positions from frame $i$ to $i-1$. Eq. (9) is then applied to fill the remaining regions. As illustrated in the Figure 3(c), our cross-frame alignment strategy enables seamless sampling propagation across frames and artifact reduction.

## 4.3 LATENT SPACE CORRECTION

After the latent space permutation and resampling, the original latent $Z_{T_0}^{1:N}$ is updated to $Z_{T_0}^{'1:N}$. As shown in Figure 4(a), continuing the denoising process with the updated latent as usual leads to the poor-quality results. Inspired by Hwang et al. (2024), we analyze this issue through the signal-to-noise ratio (SNR). The SNR of the noised latent is defined as $\bar{\alpha}_t/(1-\bar{\alpha}_t)$ in Eq. (1) during training, and the model implicitly expects corresponding SNR at each timestep during inference. We hypothesize that permutation and resampling alter the SNR of latent, creating a mismatch between training and inference, thereby degrading the output quality.

However, the SNR of the updated latent cannot be directly estimated due to random sampling. Since the base model, Animatediff Guo et al. (2024), is trained with noise supervision, we are inspired to ask: *Can the SNR of the input latent be inferred from the variance of the predicted noise?* To investigate this, we analyze the standard denoising process. As shown in Figure 4(c), as the theoretical SNR of the input latent increases, the variance of the predicted noise decreases accordingly. Notably, this variance remains highly consistent across different random seeds, indicating that the model produces noise with a characteristic variance for each SNR value. Therefore, we can indirectly estimate SNR changes by monitoring the variance of the predicted noise.

Based on this observation, we further analyze how the update operation affects the SNR of the latent by examining the variance of the predicted noise. As shown in Figure 4(d), at timestep $T_0 = 25$, the updated latent exhibits a significantly higher noise variance, indicating an SNR shift that disrupts subsequent denoising. Moreover, this effect varies with camera motion: the more new perspectives are introduced, the greater the SNR shift. Such a shift creates a mismatch between training and inference, impairing the model's ability to predict accurate noise during the denoising process.

To mitigate the SNR shift analyzed above, we reintroduce noise into the updated latent during denoising for unified correction. Specifically, after denoising to $Z_{T_1}^{'1:N}$, we inject noise as defined in Eq. (1) to obtain $Z_{T_2}^{*1:N}$, then resume denoising from $T_2$ to 1. As shown in Figure 4 (d), the predicted noise variance of the renoised latent closely aligns with that of the original latent, indicating that this correction strategy effectively mitigates the SNR shift. Figure 4(b) further demonstrates that this approach significantly enhances video generation quality across different camera parameters.

| Methods | Generation Quality | | | Camera Controllability | | | User Study | | Tuning-free |
|---|---|---|---|---|---|---|---|---|---|
| | FVD↓ | CLIP-F↑ | CLIP-T↑ | Pan.-Error↓ | Zoom.-Error↓ | Rot.-Error↓ | Quality↑ | Controllability↑ | |
| Animatediff Guo et al. (2024) | 5645.6 | 98.28 | 32.16 | 0.0873 | 0.1913 | - | 79.93 | 86.37 | ✗ |
| Direct-A-Video Yang et al. (2024) | 5340.2 | 97.87 | 31.16 | 0.3447 | 0.1742 | - | 60.53 | 59.89 | ✗ |
| CameraCtrl He et al. (2025) | 5495.6 | 98.56 | 33.48 | 0.1896 | 0.2534 | 0.4211 | 48.56 | 55.29 | ✗ |
| MotionCtrl Wang et al. (2024d) | 5498.2 | 98.59 | 31.19 | 0.2298 | 0.1599 | 0.2871 | 79.88 | 88.39 | ✗ |
| MotionBooth Wu et al. (2024) | 5505.3 | 97.86 | 33.18 | 0.1386 | - | - | 64.50 | 66.47 | ✓ |
| LightMotion (Ours) | **5329.5** | **98.62** | **33.56** | **0.0532** | **0.1590** | **0.2351** | **88.55** | **88.70** | ✓ |

Table 1: Quantitative comparison on the full dataset. Best result in **bold**, second-best underlined.

## 5 EXPERIMENTS

### 5.1 EXPERIMENTAL SETTINGS

**Implementation Details.** LightMotion builds on a widely-used T2V framework[4] Animatediff-V2 Guo et al. (2024), which is trained on 16-frame sequences at a resolution of $512 \times 512$. The hyper-parameters for LightMotion are as follows: $T_0 = 0.5T$, $T_1 = 1$, $T_2 = 0.7T$, and $T = 50$. LightMotion can generate videos with camera motion on one RTX 3090 GPU, without fine-tuning.

**Datasets Details.** Following prior works Wu et al. (2024); Hou et al. (2024), we construct a comprehensive dataset of 1,600 prompt-motion pairs, combining 100 diverse prompts with 16 distinct camera motions each. The dataset includes diverse subjects and scenes to support video evaluation.

**Baselines.** We compare our LightMotion with five camera motion control baselines: Animatediff with LoRA Guo et al. (2024), Direct-A-Video Yang et al. (2024), CameraCtrl He et al. (2025), MotionCtrl Wang et al. (2024d) and MotionBooth Wu et al. (2024). Note that Motionbooth and our LightMotion are tuning-free, while other approaches need fine-tuning.

### 5.2 QUANTITATIVE COMPARISON

We evaluate the quality of generated videos with camera motion across four aspects: *generation quality*, *camera controllability*, *user study*, and *GPT-4o evaluation*, as follow:

**Generation Quality.** To evaluate the generation quality, we use three metrics: Fréchet Video Distance (FVD) Unterthiner et al. (2018) in MSR-VTT Xu et al. (2016), CLIP text-image similarity (CLIP-T) Radford et al. (2021), and CLIP frame-by-frame similarity (CLIP-F) Radford et al. (2021). These metrics collectively measure the visual quality, text-video alignment, and cross-frame coherence. As shown in Table 1, LightMotion outperforms all baselines in FVD, CLIP-T, and CLIP-F, showing its advantages in visual quality, text-video alignment, and cross-frame coherence.

**Camera Controllability.** Camera controllability is assessed by errors in panning, zooming, and rotation Shi et al. (2023); Wang et al. (2024a), referred as Pan.-Error, Zoom.-Error, and Rot.-Error. All methods are evaluated based on the camera motions they support. As shown in Table 1, our LightMotion supports all three motions and achieves the best performance on Pan.-Error, Zoom.-Error, and Rot.-Error, showing its outstanding camera controllability in a tunning-free manner.

**User Study.** In addition, we invite 100 participants to score (max 100) the quality and controllability of videos generated by different methods. As shown in Table 1, our LightMotion outperforms other baselines in both quality and controllability based on user preferences.

**GPT-4o Evaluation.** Furthermore, we utilize the GPT-4o Achiam et al. (2023) to assess video quality, content coherence, and camera controllability of generated videos. In Table 2, our

| Methods | Quality↑ | Coherence↑ | Controllability↑ | Avg↑ |
|---|---|---|---|---|
| Animatediff Guo et al. (2024) | 91.0 | 85.3 | 85.0 | 87.1 |
| Direct-A-Video Yang et al. (2024) | 85.8 | 84.7 | 74.8 | 81.8 |
| CameraCtrl He et al. (2025) | 86.1 | 83.1 | 80.1 | 83.1 |
| MotionCtrl Wang et al. (2024d) | 90.8 | 85.8 | **87.2** | 87.9 |
| MotionBooth Wu et al. (2024) | 82.5 | 82.5 | 70.0 | 78.3 |
| LightMotion (Ours) | **92.1** | **85.9** | 86.3 | **88.1** |

Table 2: Overall performance comparison evaluated by GPT-4o Achiam et al. (2023). Best scores are in **bold**, second-best are underlined.

LightMotion achieves best in quality and coherence, slightly behind in controllability. Overall, LightMotion delivers the superior average performance, further demonstrating its advantage.

---

[4]See the **supplementary materials** for other frameworks.

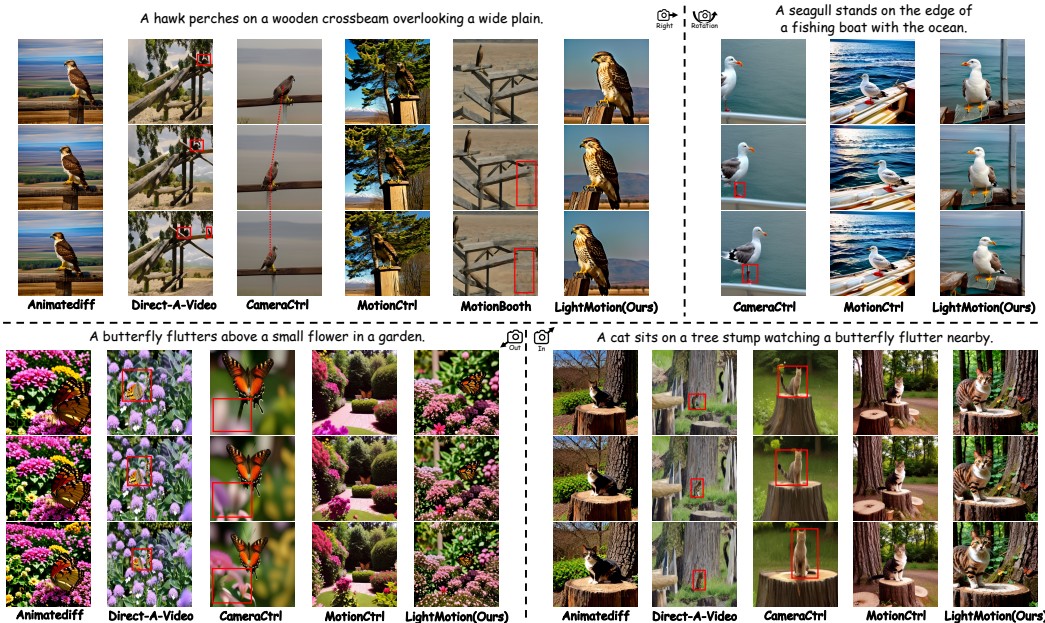

Figure 5: Qualitative comparisons across baselines, showing only supporting methods per motion.

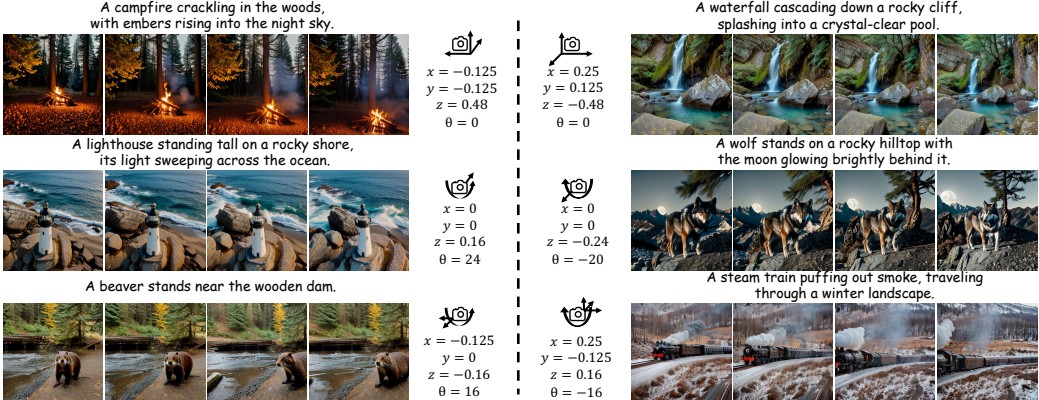

Figure 6: LightMotion simulates camera motion in video generation without fine-tuning. Combining three basic motions: panning, zooming, and rotation, it can generalize to various camera movements.

## 5.3 QUALITATIVE COMPARISON

We present qualitative comparisons in Figure 5, displaying only the camera motions supported by each method. The results highlight our LightMotion's superior performance across all camera motions, outperforming existing methods in generation quality, controllability, and coherence. *For camera panning*, MotionBooth Wu et al. (2024) and Direct-A-Video Yang et al. (2024) suffer from object repetition, while CameraCtrl He et al. (2025) is insensitive to long-distance movements. *For camera zooming*, Direct-A-Video introduces artifacts that distort the object's semantics, while CameraCtrl causes inconsistencies between frames. *For camera rotation*, CameraCtrl suffers from significant degradation in visual quality. **Additionally**, the combination of these three fundamental camera motions can cover almost real-world camera movements. As illustrated in Figure 6, given the user-defined text prompt and camera parameters, LightMotion can generate coherent video sequences with realistic camera motion without additional fine-tuning.

## 5.4 ABLATION STUDIES

**Ablation Study on Hyper-Parameters.** We investigate the setting of three hyper-parameters: $T_0$, $T_1$, and $T_2$. First, the grid search is used to explore the optimal combination, the timestep $T_0$

Figure 7: The ablation study of different hyper-parameters $T_0$, $T_2$ (left) and $T_1$ (right). Both the updated timestep $T_0$ and the correction timestep $T_2$ significantly influence overall performance. Introducing noise early disrupts semantics and motions. Best viewed with zooming-in.

for update operation and the timestep $T_2$ for correction mechanism, with the details provided in Figure 7(left). Early update operation has a low signal-to-noise ratio and causes artifacts, while later ones offer richer semantics but may lead to mismatches in new perspectives. In terms of latent space correction, a short noised timestep fails to correct the SNR shift, while a longer noised step increases inference time. Furthermore, we investigate the impact of introducing noise at different timestep $T_1$, with the details shown in Figure 7(right). Early noise injection will disrupt the object's semantics and yields minimal motions simulated by permutation. To match the training process, we set $T_1 = 1$ to first obtain nearly clean $Z_1'^{1:N}$ and then perform the subsequent correction process.

**Ablation Study on Core Components.** Additionally, we investigate three core components proposed in our LightMotion: *background-aware sampling*, *cross-frame alignment*, and *latent space correction*, validating them across eight variants in Table 3. Due to the limitations of computing resources, each variant is validated on the half of dataset. In particular, the background-aware sampling and cross-frame alignment strategies collaborate to significantly enhance both generation quality and coherence across frames. Meanwhile, the latent space correction mechanism greatly enhances

| Components | | | Quality | | | Controllability | | |
|---|---|---|---|---|---|---|---|---|
| B.A.S. | C.F.A. | L.S.C. | FVD ↓ | CLIP-F ↑ | CLIP-T ↑ | Pan.-Error ↓ | Zoom.-Error ↓ | Rot.-Error ↓ |
| ✗ | ✗ | ✗ | 5198.9 | 96.43 | 32.58 | 0.0305 | 0.3078 | 0.7619 |
| ✓ | ✗ | ✗ | 4937.7 | 96.47 | 32.64 | 0.0256 | 0.2813 | 0.7821 |
| ✗ | ✓ | ✗ | 5181.6 | 96.62 | 32.59 | 0.0346 | 0.3272 | 0.7939 |
| ✗ | ✗ | ✓ | 6157.9 | 98.35 | 33.20 | 0.0548 | 0.1878 | 0.2461 |
| ✓ | ✓ | ✗ | **4907.5** | 96.75 | 32.65 | 0.0232 | 0.2577 | 0.8021 |
| ✓ | ✗ | ✓ | 5417.2 | 98.34 | 33.23 | 0.0234 | 0.1844 | 0.2432 |
| ✗ | ✓ | ✓ | 5751.2 | **98.36** | 33.21 | 0.0305 | 0.1828 | 0.2379 |
| ✓ | ✓ | ✓ | 5226.1 | **98.36** | **33.27** | **0.0179** | **0.1756** | **0.2357** |

Table 3: Ablation study of the three core components on half of the dataset. Best results are shown in **bold**. Abbreviations: "B.A.S." is background-aware sampling; "C.F.A." is cross-frame alignment; "L.S.C." is latent space correction.

camera controllability with only a minor trade-off in FVD. Overall, the synergy of these components effectively enhances stability and achieves an optimal balance in performance.

# 6 LIMITATIONS AND FUTURE WORK

Our LightMotion also has some limitations: (i) The Latent space permutation and resampling strategies perform poorly in cases involving rapid camera motion with numerous new perspectives. (ii) While effective, the latent space correction mechanism incurs additional inference time.

In this way, more effective strategies for modeling high-speed camera movements and more efficient mechanisms for correcting the SNR shift can be explored in future work.

# 7 CONCLUSION

We propose LightMotion, a light and tuning-free approach that enhances video generation with camera motion. LightMotion operates in the latent space, eliminating the need for inpainting or depth estimation, achieving end-to-end inference. Our contributions include: (1) Latent space permutation effectively simulates general camera motions. (2) Latent space resampling incorporates background-aware sampling and cross-frame alignment, filling new perspectives while maintaining frame consistency. (3) Latent space correction mitigates the SNR shift caused by permutation and resampling, enhancing video quality. Exhaustive experiments show that our method surpasses existing methods in both quantitative metrics and qualitative evaluations.

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

## A    LLM USAGE STATEMENT

A large language model (LLM) was used for language polishing of the manuscript. In addition, the LLM was employed to assist in evaluating the generation quality, temporal coherence, and camera controllability of the generated videos, as reported in Table 2 of the main paper.

## B    DETAILED EXPERIMENTAL SETTINGS

### B.1    CAMERA PARAMETER DEFINITIONS

Unlike traditional camera-controlled video generation models, our LightMotion eliminates the need for users to input technical camera parameters such as intrinsic, rotation, or translation matrices. Instead, we streamline the input parameters without requiring knowledge of camera geometry, thus lowering the barrier for non-professional users. Specifically, We only require users to input four camera parameters: $x, y, z$, and $\theta$. By combining these parameters, we can simulate various camera motions in the real world.

Following previous work, Direct-A-Video Yang et al. (2024), we define the parameters as follows: $x$ represents the $X$-pan ratio, defined as the total horizontal shift of the frame center from the first to the last frame related to the frame width, with $x > 0$ indicating the panning rightward. $y$ denotes the $Y$-pan ratio, which indicates the total vertical shift of the frame center related to the frame height, with $y > 0$ indicating the panning downward. $z$ refers to the $Z$-pan zooming ratio, defined as the scaling factor between the first and last frame, with $z > 0$ indicating zooming-in.

Different from Direct-A-Video, we additionally model the camera rotation and define relative parameters. We model rotation using point cloud projection theory, which primarily involves camera intrinsic parameters $K$, rotation matrices $R^i$, and depth information $d(u, v, 1)$. Here, we ignore $d(u, v, 1)$ (which will be discussed in the following section) and only consider the settings of $K$ and $R^i$ (rotation about the Y-Axis, the same applies to other cases):

$$K = \left( \begin{array}{ccc} f_x & 0 & u_0 \\ 0 & f_y & v_0 \\ 0 & 0 & 1 \end{array} \right), R^i = \left( \begin{array}{ccc} \cos \gamma^i & 0 & \sin \gamma^i \\ 0 & 1 & 0 \\ -\sin \gamma^i & 0 & \cos \gamma^i \end{array} \right). \tag{11}$$

Similar to pixel space, the camera's optical center $(u_0, v_0)$ are positioned at the center of the latent space, $(\frac{h}{2}, \frac{w}{2})$. However, the focal lengths $(f_x, f_y)$ in the latent space do not have physical significance. Through extensive experimentation, we found that $f_x = f_y = 15$ yields effective results in the latent space. Regarding the rotation matrix $R^i$ for the $i$-th frame, relative angles $\gamma^i$ are defined as $\frac{2 \cdot \theta}{N} \cdot (i - N)$, with $\gamma^i$ ranging from $-\theta$ to $\theta$ across $N$ frames. Here, $\theta$ is the user-defined rotation parameter, and $\theta > 0$ indicates counterclockwise rotation.

### B.2    CAMERA PARAMETER SETTINGS

Since not all methods support every type of camera motion, we define 16 distinct camera motion scenarios, including 8 panning, 4 zooming, and 4 rotation sequences, to assess the performance of each model on the respective motion types. Following, we will provide a detailed description of the parameter settings for these camera motions.

For panning, we define 4 motion types, including leftward, rightward, upward, and downward movements, each with two variations: small-scale and large-scale. Small-scale panning shifts the frame from first to last, covering 25% of the frame width, while large-scale panning covers 50%. Parameter settings are detailed in Table 4.

For zooming, we define two motion types: zooming-in and zooming-out, each having small-scale and large-scale variations. Small-scale zooming scales the frame from first to last, covering 24% of the frame size, while large-scale spanning 48%. Parameter settings are detailed in Table 5.

For rotation, we also define two motion types: counterclockwise rotation and clockwise rotation, each having small-scale and large-scale variations. Small-scale rotation rotates the frame from first to last, ranging from $-\theta$ to $\theta$ where $\theta = 8$, while large-scale rotation uses $\theta = 16$. Parameter settings are detailed in Table 6.

| Camera Motion | Parameter Settings |
|---|---|
| Leftward (small-scale) | $x = -0.25, y = 0.00$ |
| Leftward (large-scale) | $x = -0.50, y = 0.00$ |
| Rightward (small-scale) | $x = 0.25, y = 0.00$ |
| Rightward (large-scale) | $x = 0.50, y = 0.00$ |
| Upward (small-scale) | $x = 0.00, y = -0.25$ |
| Upward (large-scale) | $x = 0.00, y = -0.50$ |
| Downward (small-scale) | $x = 0.00, y = 0.25$ |
| Downward (large-scale) | $x = 0.00, y = 0.50$ |

Table 4: Camera panning parameter settings.

| Camera Motion | Parameter Settings |
|---|---|
| Zooming-in (small-scale) | $z = 0.24$ |
| Zooming-in (large-scale) | $z = 0.48$ |
| Zooming-out (small-scale) | $z = -0.24$ |
| Zooming-out (large-scale) | $z = -0.48$ |

Table 5: Camera zooming parameter settings.

| Camera Motion | Parameter Settings |
|---|---|
| CCW. rotation (small-scale) | $\theta = 8$ |
| CCW. rotation (large-scale) | $\theta = 16$ |
| CW. rotation (small-scale) | $\theta = -8$ |
| CW. rotation (large-scale) | $\theta = -16$ |

Table 6: Camera rotation parameter settings. "CCW." represents the counterclockwise while "CW." renotes the clockwise.

## C  RELATED PROOFS

**Theorem 1.** Rotation of a 3D point cloud along the x, y, or z axis is inherently independent of depth.

*Proof.* Let $(u, v, 1)^T$ be the pixel coordinates in the original latent space, $K$ the camera intrinsic matrix, and $(X_c, Y_c, d(u, v, 1))^T$ the spatial coordinates after point cloud projection, with $d(u, v, 1)$ representing the depth. Through the pin-hole camera model, we have:

$$d(u,v,1) \begin{pmatrix} u \\ v \\ 1 \end{pmatrix} = K \cdot \begin{pmatrix} X_c \\ Y_c \\ d(u,v,1) \end{pmatrix} = \begin{pmatrix} f_x & 0 & c_x \\ 0 & f_y & c_y \\ 0 & 0 & 1 \end{pmatrix} \cdot \begin{pmatrix} X_c \\ Y_c \\ d(u,v,1) \end{pmatrix}, \quad (12)$$

where $f_x$ and $f_y$ are the focal lengths, and $c_x$ and $c_y$ are the coordinates of the camera's optical center.

Rearranging the above equations, we obtain:

$$\begin{cases} u = f_x \cdot \frac{X_c}{d(u,v,1)} + c_x \\ v = f_y \cdot \frac{Y_c}{d(u,v,1)} + c_y \end{cases} \Rightarrow \begin{cases} X_c = \frac{(u - c_x)}{f_x} \cdot d(u,v,1) \\ Y_c = \frac{(v - c_y)}{f_y} \cdot d(u,v,1) \end{cases}. \quad (13)$$

According to the point cloud projection theory, by rotating the point cloud to another perspective using the rotation matrix $R_y$ (taking the rotation around the $Y$-axis as an example, with the same principle applying to rotations around other axes), we obtain the following equation:

$$\begin{pmatrix} X' \\ Y' \\ Z' \end{pmatrix} = R_y \cdot \begin{pmatrix} X_c \\ Y_c \\ d(u,v,1) \end{pmatrix} = \begin{pmatrix} \cos\theta & 0 & \sin\theta \\ 0 & 1 & 0 \\ -\sin\theta & 0 & \cos\theta \end{pmatrix} \cdot \begin{pmatrix} X_c \\ Y_c \\ d(u,v,1) \end{pmatrix}. \quad (14)$$

Substituting Eq. (13) and simplifying, we have:

$$\left( \begin{array}{c} X' \\ Y' \\ Z' \end{array} \right) = \left( \begin{array}{c} \cos\theta \cdot X_c + \sin\theta \cdot d(u,v,1) \\ Y_c \\ -\sin\theta \cdot X_c + \cos\theta \cdot d(u,v,1) \end{array} \right) = \left( \begin{array}{c} \frac{\cos\theta \cdot d(u,v,1)\cdot(u-c_x)}{f_x} + \sin\theta \cdot d(u,v,1) \\ \frac{(v-c_y)}{f_y} \cdot d(u,v,1) \\ \frac{-\sin\theta \cdot d(u,v,1)\cdot(v-(y)}{f_y} + \cos\theta \cdot d(u,v,1) \end{array} \right).$$

(15)

Then, we can derive the following ratio relationship:

$$\begin{cases} \frac{X'}{Z'} = \frac{\cos\theta \cdot f_y \cdot (u-c_x) + \sin\theta \cdot f_x \cdot f_y}{-\sin\theta \cdot f_x \cdot (v-c_y) + \cos\theta \cdot f_x \cdot f_y} \\ \frac{Y'}{Z'} = \frac{(v-c_y)}{-\sin\theta \cdot (v-c_y) + \cos\theta \cdot f_y} \end{cases}.$$

(16)

On the other hand, the point cloud in the new perspective can be mapped to the new pixel coordinates $(u', v', 1)$ as in Eq. (12), satisfying the following relationship:

$$Z' \left( \begin{array}{c} u' \\ v' \\ 1 \end{array} \right) = K \cdot \left( \begin{array}{c} X' \\ Y' \\ Z' \end{array} \right) = \left( \begin{array}{ccc} f_x & 0 & C_x \\ 0 & f_y & C_y \\ 0 & 0 & 1 \end{array} \right) \cdot \left( \begin{array}{c} X' \\ Y' \\ Z' \end{array} \right).$$

(17)

Substituting Eq. (16) and simplifying, we obtain:

$$\begin{cases} u' = f_x \cdot \frac{X'}{Z'} + c_x = \frac{\cos\theta \cdot f_y (u-c_x) + \sin\theta f_x \cdot f_y}{-\sin\theta \cdot (v-c_y) + \cos\theta \cdot f_y} + c_x \\ v' = f_y \cdot \frac{Y'}{Z'} + c_y = \frac{f_y \cdot (v-c_y)}{-\sin\theta(v-c_y) + \cos\theta \cdot f_y} + c_y \end{cases}.$$

(18)

The results show that projected pixel coordinates are independent of depth information $d(u,v,1)$.

*Proof End.*

**Theorem 2.** By considering panning along the x and y axes, zooming along the z axis, and rotations around the x, y, and z axes, these six basic motions can approximate nearly all general camera movements in real-world scenarios.

*Proof.* Given an any camera pose $[R \mid T]$, we construct the rotation matrix $R$ and translation matrix $T$ in sequence.

*(i) Rotation matrix.* According to the Euler angle formulation, $R$ can be decomposed into successive rotations along the $x$, $y$, and $z$ axes. This process can be formulated as:

$$R = R_x(\alpha) \cdot R_y(\beta) \cdot R_z(\gamma),$$

(19)

where $\alpha$, $\beta$, and $\gamma$ are the Euler angles, which can be efficiently computed using an Euler angle solver. The matrices $R_x(\alpha)$, $R_y(\beta)$, and $R_z(\gamma)$ are identical to those used in our point cloud formulation.

*(i) Translation matrix.* Due to the linear additivity of translation, $T$ can be decomposed into independent translations along the $x$, $y$, and $z$ axes. This can be expressed as:

$$T = T_x(x) + T_y(y) + T_z(z),$$

(20)

where $T_x(x)$ and $T_y(y)$ can be covered by our panning mapping function $\mathcal{F}_{paning}$ along the $x$ and $y$ axes, respectively, while $T_z(z)$ can be covered by our zooming mapping function $\mathcal{F}_{zooming}$ along the $z$ axis.

*Proof End.*

## D  COMPATIBILITY WITH DIFFERENT FRAMEWORK

Besides Animatediff-V2 Guo et al. (2024), our method can also be integrated into other video generation frameworks such as Animatediff-V3 and LaVie Wang et al. (2024c) at a resolution of $512\times512$, and into ZeroScope Sterling (2023) with a different resolution of $320 \times 576$. Additional qualitative results are presented in Figure 8 and Figure 9.

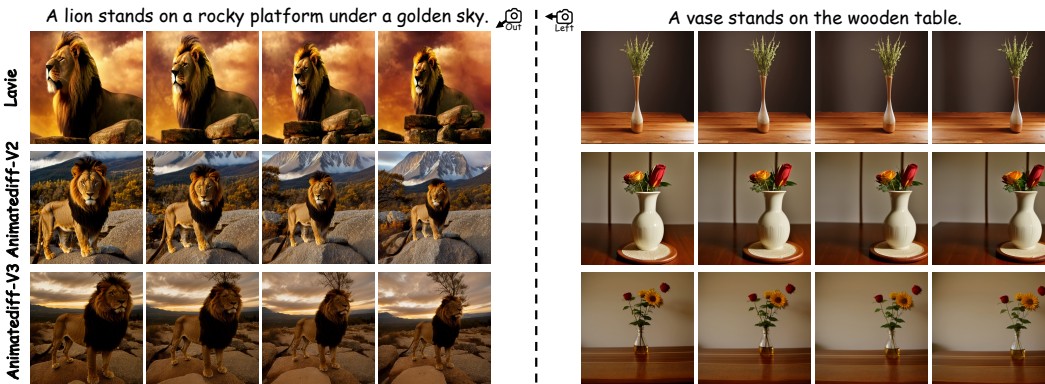

Figure 8: Our LightMotion can be seamlessly integrated into most existing frameworks

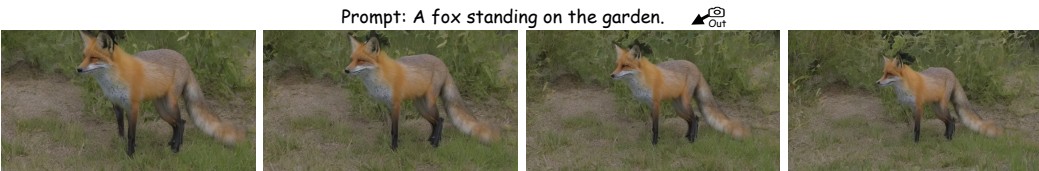

Figure 9: Our LightMotion can be integrated into ZeroScope with a resolution of $320 \times 576$.

# E COMPATIBILITY WITH NON-LINEAR MOVEMENTS

In addition to the linear motions demonstrated in the main text, LightMotion also supports non-linear movements across frames by adjusting the coordinate mapping functions, as illustrated in Figure 10.

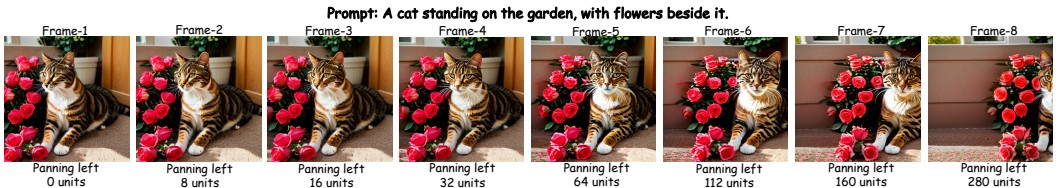

Figure 10: LightMotion supports non-linear movements by adjusting coordinate mappings.

# F ADDITIONAL EXAMPLES WITH OBJECT REPETITION

Building on the cases presented in the main paper, Figure 11 provides further examples that highlight the widespread nature of object repetition and inter-frame inconsistency.

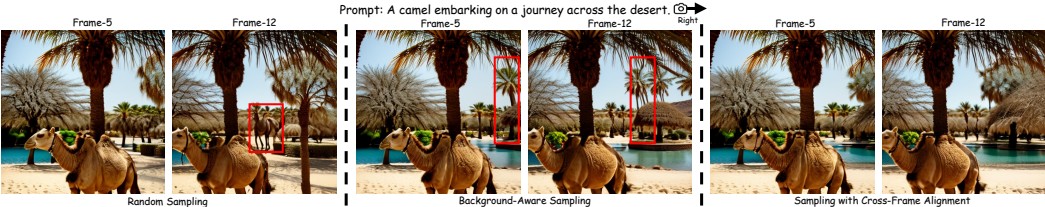

Figure 11: Additional examples of object repetition and inter-frame inconsistency.

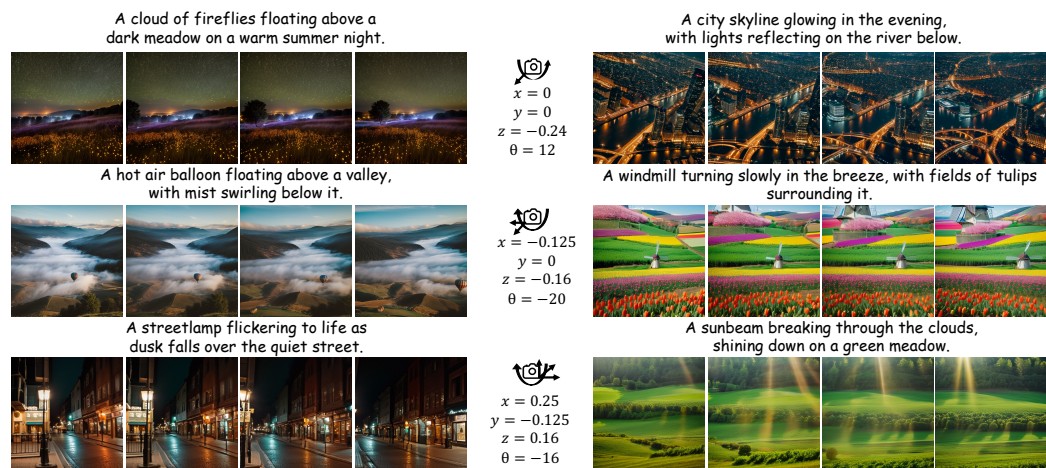

Figure 12: Additional visualization with camera motion using user-defined parameter combinations.

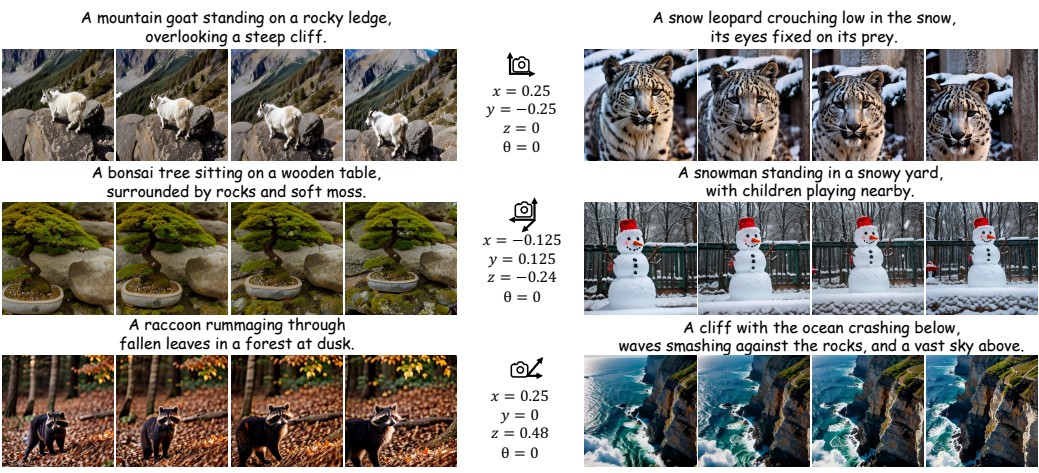

Figure 13: Additional visualization with camera motion using user-defined parameter combinations.

## G VARIOUS CAMERA COMBINATIONS

Furthermore, our LightMotion supports a wide variety of camera combinations, with additional visual results provided in Figure 12 and Figure 13, demonstrating its versatility.

## H ADDITIONAL QUALITATIVE COMPARISON

To highlight the superiority of LightMotion, we provide additional qualitative comparisons in Figure 14 and Figure 15, showing its performance across various camera motions.

## I USER STUDY

We further assess the preferences of users for various methods through a user study. Specifically, we design a questionnaire that includes 10 sets of videos generated by different methods. These sets include two groups featuring camera panning, four groups focusing on camera zooming, and four groups highlighting camera rotation. Additionally, each generated video is accompanied by a relevant text description and the corresponding camera motion. In each set, the results from all methods are transformed into .gif files and presented on same page. We establish clear evaluation criteria for users, who score the videos on a scale of up to 100 in each set based on the following two

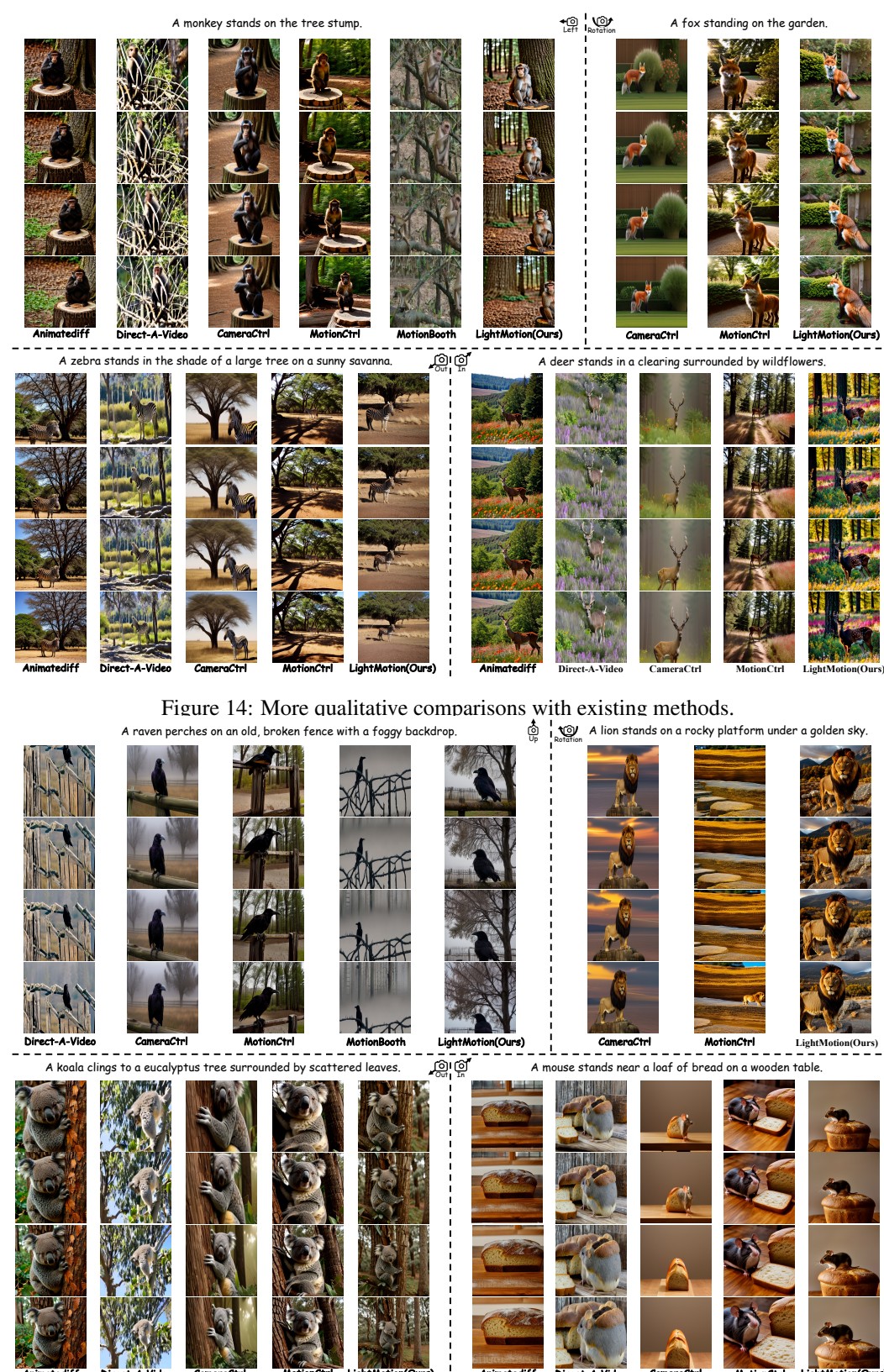

Figure 14: More qualitative comparisons with existing methods.

Figure 15: More qualitative comparisons with existing methods.

aspects: (i) *Generation Quality*: This criterion evaluates the similarity between the generated video and its text description, as well as the aesthetic quality. (ii) *Camera Controllability*: This criterion assesses the alignment between the camera movements in the generated video and the specified camera motions. To ensure fairness, the names of all methods will be concealed, and the order of the generated results in each set will be randomized. Finally, 100 valid questionnaires were included in the analysis to evaluate user preferences.

## J  GPT-4O EVALUATION

Additionally, the generated video samples for the user study will also be re-evaluated by GPT-4o. Similarly, we establish clear evaluation criteria for GPT-4o, which scores the videos on a scale of up to 100 in each set based on the following three aspects: (i) *Generation Quality*: This criterion evaluates the similarity between the generated video and its text description, as well as the aesthetic quality. (ii) *Coherence*: This criterion evaluates the semantic coherence between frames in the generated video. (iii) *Camera Controllability*: This criterion assesses the alignment between the camera movements in the generated video and the specified camera motions. To ensure robustness, we perform five repetitions and average the scores for each method.

