# OpenReview forum: "LightMotion: A Light and Tuning-free Method for Simulating Camera Motion in Video Generation"
_ICLR.cc/2026/Conference — ICLR 2026 Conference Withdrawn Submission_

### Official Review · Reviewer_yJek · 2025-10-30

**Soundness:** 3
**Presentation:** 2
**Contribution:** 2
**Rating:** 4
**Confidence:** 4

**Summary:**

This paper introduces "LightMotion," which is a simple, "tuning-free" way to add camera motion (like panning, zooming, rotating) to generated videos.

It works in the latent space, mostly by shuffling things around (permutation operations). It also has a "background-aware" resampling trick and cross-frame alignment to keep the video from looking weird or artifact-y.

A key part of the paper is that the authors found that these latent manipulations mess up the Signal-to-Noise Ratio (SNR), which hurts video quality. They figured out why this happens and proposed a fix for it. They test their method against a bunch of others with both metrics and visual comparisons.

**Strengths:**

* **It's Simple and Cheap:** I really like that this method is "tuning-free." You don't need to fine-tune a massive model, and it doesn't rely on extra, heavy models for things like depth or inpainting. It just works in the latent space, which makes it super accessible and compute-friendly.
* **Smart Resampling:** The background-aware sampling and cross-frame alignment seem to really work. Figure 3 shows it avoids those weird repeating objects and temporal artifacts that have plagued other tuning-free methods.
* **Finding (and Fixing) the SNR Shift:** This was a really cool insight. Spotting that the latent operations were causing an SNR shift—and then actually fixing it—is a solid contribution. Figure 4 does a good job showing how much this fix improves the final video quality.

**Weaknesses:**

1.  **Needs a Better Failure Analysis:** The "Limitations" section is pretty vague. It just says performance drops with "rapid, complex" motion. But what does that *mean*? Is there a specific speed or angle where it just breaks? The paper really needs a proper analysis of its failure cases. Figure 9 (ablation) hints at this, but it's not detailed enough.
2.  **The User Study is a Bit "Meh":** The user studies (Table 1, Appendix) and the GPT-4o evaluation feel a bit weak. You don't give enough detail on how they were run. Was it randomized? What did the interface look like? Were the results even statistically significant? Using GPT-4o scores just makes it feel even less transparent than just asking people.
3.  **How "General" is the Motion?** You *claim* it supports non-linear camera movements (like in Appendix E), but you're super sparse on the details. How do you actually parameterize or implement a complex, curvy camera path? The paper doesn't convincingly show that it can generalize beyond the simple pan/zoom/rotate.

**Questions:**

1.  Can you show some quantitative results or just a clearer failure analysis for extreme camera movements (like huge perspective shifts or super-fast, jerky paths)?
2.  For the user/GPT-4o study: What did you do to prevent bias (like randomization)? And did you run any statistical significance tests on the results?
3. Can you provide more experiments on more SOTA models like WAN?

---

### Official Review · Reviewer_rUmJ · 2025-10-30

**Soundness:** 2
**Presentation:** 3
**Contribution:** 2
**Rating:** 2
**Confidence:** 4

**Summary:**

The paper presents LightMotion, a lightweight and tuning-free method for simulating camera motion in video generation. Operating in the latent space, it eliminates the need for fine-tuning, inpainting, or depth estimation by introducing three components: latent space permutation for modeling panning, zooming, and rotation; latent space resampling for background-aware and temporally consistent frame synthesis; and latent space correction to mitigate SNR shifts and enhance visual quality. Experiments demonstrate that LightMotion achieves better video quality and camera controllability compared to some existing methods while maintaining high efficiency and full compatibility with several U-Net-based video diffusion models.

**Strengths:**

1. This paper is well-written and structured, making it accessible to readers with varying levels of expertise in the field.
2. Visual results seem promising compared to vanilla results yielded by AnimateDiff and some baseline methods.
3. The paper provides thorough experimental evaluations, including ablation studies on key components.

**Weaknesses:**

1. The authors do not adapt their method to (or even discuss their method on) advanced DiT-based video models such as HunyuanVideo and the WAN series. Moreover, their approach relies on cross-attention maps, which may be difficult to transfer to modern architectures that predominantly employ full 3D attention mechanisms. In addition, I believe the proposed Cross-Frame Alignment strategy would also be challenging to apply to more advanced video generation models, since these models often adopt the causal VAE, where multiple frames in pixel space are encoded into a single latent frame, making frame-by-frame region replication infeasible. These limitations raise concerns about the generalization ability of LightMotion.
2. The author should discuss (or compare with, if possible) more existing outstanding works on motion customization (both U-Net-based ones and DiT-based ones), including but not limited to:
    1. MOFT: Video Diffusion Models are Training-free Motion Interpreter and Controller
    2. MotionClone: Training-Free Motion Cloning for Controllable Video Generation
    3. VMC: Video Motion Customization using Temporal Attention Adaption for Text-to-Video Diffusion Models
    4. VD3D: Taming Large Video Diffusion Transformers for 3D Camera Control
3. This paper does not specify or thoroughly discuss the evaluation dataset since it is a self-collected one, which may not provide a comprehensive view of LightMotion’s effectiveness. Releasing more details of the evaluation dataset or incorporating videos from publicly available benchmarks would greatly enhance the credibility of the paper.

**Questions:**

Please see the weakness.

---

### Official Review · Reviewer_EWhG · 2025-10-31

**Soundness:** 3
**Presentation:** 3
**Contribution:** 3
**Rating:** 4
**Confidence:** 4

**Summary:**

This paper proposes a camera control algorithm for video generation models to enable controllable camera motion during generation.

**Strengths:**

The proposed method is training-free, making it easily integrable into any existing model.

**Weaknesses:**

The paper should include comparative experiments against other baseline models, such as the camera-control variant of Wan2.1-Fun: https://huggingface.co/alibaba-pai/Wan2.1-Fun-V1.1-14B-Control-Camera .

The method modifies the model’s forward computation, which may introduce a potential train-inference inconsistency. The authors should provide additional experiments to verify that this modification does not degrade the model’s original generative capabilities.

The generation results are not adequately demonstrated. The authors are encouraged to submit supplementary materials in video format.

**Questions:**

Please refer to the weaknesses.

---

### Official Review · Reviewer_TMGm · 2025-11-01

**Soundness:** 3
**Presentation:** 4
**Contribution:** 3
**Rating:** 6
**Confidence:** 3

**Summary:**

This paper introduces LightMotion, a lightweight and tuning-free method for simulating camera motion in video generation. The method operates directly in the latent space of pre-trained video diffusion models and requires no fine-tuning. Its core contributions are threefold: 1) Latent space permutatio which simulates fundamental camera motions (pan, zoom, rotate) through geometric transformations; 2) Latent space resampling, which uses background-aware sampling and cross-frame alignment to fill new regions coherently and avoid artifacts; 3) Latent space correction, a proposed mechanism to mitigate an identified SNR shift caused by the previous operations, thereby improving output quality. Extensive experiments show that LightMotion achieves performance comparable to tuning-based methods while being more efficient.

**Strengths:**

- The method is **well-motivated**, tackling a clear gap in the literature between expensive tuning-based methods and limited tuning-free alternatives. The three-component design is logical and addresses key challenges like artifact avoidance and quality preservation.
- The **experimental validation is comprehensive**, demonstrating that LightMotion, as a training-free method, achieves performance comparable to training-based methods across multiple metrics (FVD, CLIP scores, control errors) and in user studies.

**Weaknesses:**

- The **experiments are based on older U-Net-based video generation models** (e.g., AnimateDiff). The method's effectiveness has not been validated on current mainstream DiT-based video diffusion models, which limits the perception of its contemporary relevance.
- The explanation of the **SNR shift is insufficient**. The authors identify that permutation/resampling operations cause a quality drop and link it to an SNR shift, based on a correlation between predicted noise variance and theoretical SNR. However, this correlation is established for the standard denoising process. It is unclear if this premise holds after the non-standard latent manipulations performed by LightMotion. The core mechanism of the SNR shift lacks a solid theoretical explanation or an intuitive justification.

**Questions:**

- As raised in the weaknesses, could the authors comment on or demonstrate **LightMotion's compatibility and performance with modern DiT-based video diffusion models**?

---

### Note · Authors · 2025-11-13

I have read and agree with the venue's withdrawal policy on behalf of myself and my co-authors.